# FIDDLER: CPU-GPU ORCHESTRATION FOR FAST INFERENCE OF MIXTURE-OF-EXPERTS MODELS

**Keisuke Kamahori**[1]* **Tian Tang**[1,2]* **Yile Gu**[1] **Kan Zhu**[1] **Baris Kasikci**[1]

[1]University of Washington    [2]Tsinghua University

`{kamahori,tian21,yilegu,kanzhu,baris}@cs.washington.edu`

## ABSTRACT

Large Language Models (LLMs) with the Mixture-of-Experts (MoE) architectures have shown promising performance on various tasks. However, due to the huge model sizes, running them in resource-constrained environments where the GPU memory is not abundant is challenging. Some existing systems propose to use CPU resources to solve that, but they either suffer from the significant overhead of frequently moving data between CPU and GPU, or fail to consider distinct characteristics of CPUs and GPUs. This paper proposes *Fiddler*, a resource-efficient inference system for MoE models with limited GPU resources. *Fiddler* strategically utilizes CPU and GPU resources by determining the optimal execution strategy. Our evaluation shows that, unlike state-of-the-art systems that optimize for specific scenarios such as single batch inference or long prefill, *Fiddler* performs better in all scenarios. Compared against different baselines, *Fiddler* achieves 1.26 times speed up in single batch inference, 1.30 times in long prefill processing, and 11.57 times in beam search inference. The code of *Fiddler* is publicly available at `https://github.com/efeslab/fiddler`.

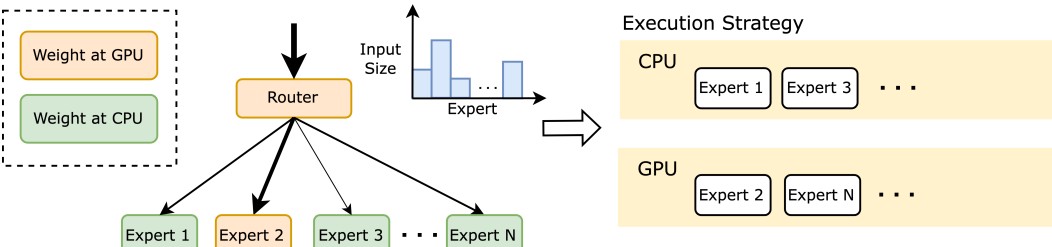

Figure 1: High level overview of *Fiddler*. Each layer of the MoE model is placed on either the CPU memory or the GPU memory, and *Fiddler* determines the optimal execution strategy using both the CPU and the GPU based on the number of input tokens of each expert.

## 1 INTRODUCTION

Recently, running Large Language Models (LLMs) in a resource-constrained environment is becoming increasingly important and relevant. There is a growing interest in running LLMs in local environments such as a personal computer or an edge device (Giacinto, 2023; Anand et al., 2023; Song et al., 2023) to improve privacy (Martínez Toro et al., 2023) and to customize models using proprietary or personal data (Lyu et al., 2023). Enabling these models to operate in resource-limited settings democratizes access to advanced LLM technologies, particularly for those without access to high-end or large numbers of GPUs. This trend is strengthened by the proposals to use LLMs at the core of all computer systems (Packer et al., 2023; Berger & Zorn, 2024). Hence, it is desirable to be able to run large models on a wide range of computers or servers, unlike the status quo where LLMs are usually served with large GPU clusters (Patel & Ahmad, 2023; Ye et al., 2025; Zhu et al., 2024).

---

*Equal contribution.

LLMs utilizing Mixture-of-Experts (MoE) architectures are particularly well suited for resource-constrained environments. MoE-based LLMs have demonstrated value across a range of tasks (Du et al., 2022; Fedus et al., 2022; Jiang et al., 2024; Databricks, 2024) and they work by selectively activating a subset of parameters via a gating mechanism, thereby lowering computational requirements for both training and inference compared to dense counterparts. Thus, MoE models are easier to scale to larger sizes, leading to the development of powerful models (Rajbhandari et al., 2022).

Although MoE models appear well-suited for resource-constrained environments due to their relatively low computational requirements, implementing local inference with these models presents several challenges. First, the model size is usually very large and scales up quickly as the number of experts or the hidden dimension increases. Recently-released MoE models include Mixtral-8x7B (47B parameters), Mixtral-8x22B (141B parameters), DBRX (132B parameters), DeepSeek-V2 (236B parameters), Grok-1 (314B parameters), and Snowflake Arctic (479B parameters) (Jiang et al., 2024; AI, 2024; Databricks, 2024; DeepSeek-AI, 2024; xAI, 2024; Snowflake, 2024). Hence, a very large number of GPUs are required just to store model parameters, given the typically limited capacity of GPU memory. The situation is further exacerbated by the virtually unlimited number of expert components in MoE models; for example, the largest variant of Switch Transformer has 2048 experts in each layer and has a total of 1.6T parameters (Fedus et al., 2022). Without model compression or quantization techniques, the model could take 3.2TB of storage. This means that 40 NVIDIA A100 80GB GPUs would be needed just to store all the model weights.

Second, while all parameters must be stored in GPU memory for efficient inference, the unique property of MoE models means that not all parameters are used to generate a new token. The fact that only a subset of parameters is active during the generation of each token leads to underutilized GPU memory. This is particularly problematic since the soaring global demand for generative AI technologies has driven GPU prices up. As a result, investing in a large number of GPUs is not cost-effective for most organizations. Only major cloud providers, known as hyperscalers, can afford it because they serve many users simultaneously and achieve better GPU utilization through significant batching.

Some existing systems use CPU resources to address the challenge of running large models in resource-limited environments, but they struggle with the efficient execution of MoE models. On the one hand, methods that offload model weights to CPU memory and transfer required weights to GPU memory on demand address memory capacity issues. However, they introduce significant runtime overhead due to the low bandwidth of the PCIe connection (Eliseev & Mazur, 2023; Xue et al., 2024b). On the other hand, CPU-based inference frameworks that partially utilize GPUs can reduce parameter transfer overhead (ggml authors, 2023). However, they fail to account for MoE model properties or different device characteristics of CPUs and GPUs, leading to suboptimal performance in critical use cases like long prefill or beam search, which is essential for enhanced generation quality (Dong et al., 2022; von Platen, 2023). We discuss shortcomings of existing approaches in detail in §2.

In this paper, we tackle the challenge of efficiently running MoE models with limited GPU resources, by strategically utilizing both CPU and GPU resources. We introduce *Fiddler*, a resource-efficient MoE inference system that intelligently leverages the heterogeneous computing architecture of both CPUs and GPUs. Unlike prior work that either only uses CPU memory or naively splits execution between CPUs and GPUs, our approach generates optimal execution strategies by considering the different characteristics of CPUs and GPUs. As CPUs have larger memory capacity despite having weaker computational power, MoE models are particularly interesting for this context due to their small computational requirement relative to their parameter size.

During inference, *Fiddler* develops a latency model based on different batching effects of CPUs and GPUs to determine the optimal execution strategy for MoE layers, as shown in Figure 1. When expert layers are executed on the CPU, latency increases almost linearly with the input size (see detailed analysis in §A). In contrast, GPU execution latency remains nearly constant regardless of input size but incurs an overhead if the weights need to be transferred from CPU memory to GPU memory. Therefore, for smaller input sizes, it is more efficient to execute expert layers on CPUs, avoiding the overhead of weight transfer. However, for larger input sizes and batch sizes, CPU computation becomes too time-consuming, making it more efficient to transfer weights to GPU memory and perform computations on the GPU. *Fiddler* dynamically chooses the execution plans that run MoE models efficiently with limited GPU memory across various workloads, including long prefill and beam search.

We also incorporate several optimizations into the design of *Fiddler*. To maximize the likelihood that the required expert is available in the GPU memory, we place frequently-used experts on the GPU based on offline profiling of expert popularity. Additionally, we design a specialized computation kernel for expert processing on the CPU using the `AVX512_BF16` instruction set, which is not supported in the native PyTorch implementation (Paszke et al., 2019).

We evaluate *Fiddler* with the uncompressed (16-bit) Mixtral-8x7B model, which has over 90GB of parameters, on two environments with single GPU each. *Fiddler* achieves on average 1.26 times speed up in single batch inference, 1.30 times in long prefill processing, and 11.57 times in beam search inference, compared against different state-of-the-art systems, across different environments (§4). Notably, while existing systems show different trade-offs (*e.g.*, offloading-based approaches excel in long prefill scenarios, while CPU-based methods perform well with single batch latency), our system integrates the advantages of both, achieving balanced and efficient results in diverse conditions.

To summarize, our contributions are as follows:

- We design *Fiddler*, an inference system for MoE models running on resource-constrained heterogeneous architectures, which finds the optimal execution strategy using both the GPU and the CPU.
- We evaluate *Fiddler* and show that it achieves better performance in single batch inference, long prefill processing, and beam search inference, compared to different state-of-the-art systems each. It shows that *Fiddler* integrates the advantages of different types of existing systems.

## 2 RELATED WORK

### 2.1 MIXTURE-OF-EXPERTS

LLMs based on MoE architecture have been showing promising performance in various applications (Rajbhandari et al., 2022; Du et al., 2022; Fedus et al., 2022; Jiang et al., 2024; Xue et al., 2024a; Dai et al., 2024). Unlike original dense Transformers (Vaswani et al., 2017), MoE models add sparsity to the feed-forward layer through a system of experts and a gating mechanism. Each MoE layer contains multiple expert layers that match the shape of the feed-forward layer, and a gating network determines which experts are activated for each input. While an MoE layer can include thousands of experts (Fedus et al., 2022), only a select few are activated by the gating network during training or inference.

### 2.2 LARGE MODELS DEPLOYMENT WITH HETEROGENEOUS ARCHITECTURE

Deploying MoE models efficiently can be challenging because of their large model size, particularly in resource-constrained settings. Some existing systems utilize CPU resources to solve the challenge of running large models in resource-limited environments, but they fall short of running MoE models efficiently.

Offloading is one approach to run large models in such an environment. They store a subset of model weights in the CPU memory instead of the GPU memory to utilize the larger capacity (Sheng et al., 2023). The required weights are transferred on demand from the CPU memory to the GPU memory during computation for inference. For MoE models, some previous works attempted to offload expert weights with caching or prefetching mechanisms (Eliseev & Mazur, 2023; Xue et al., 2024b). These approaches address memory capacity limitations and are good for throughput-oriented scenarios. However, they suffer significant latency overhead due to the frequent transfer of expert weights between the CPU and the GPU over the PCIe connection, because its bandwidth is smaller than memory access bandwidth. As a result, they show suboptimal performance for the settings where latency is critical for user experience. *Fiddler* overcomes this challenge by utilizing the computation resources of CPUs.

Another line of work proposes CPU-based inference frameworks that support running LLMs by partially using GPUs (ggml authors, 2023). Depending on the availability of GPU memory, such systems execute a subset of the model layers on the GPU and the rest on the CPU. Although they can

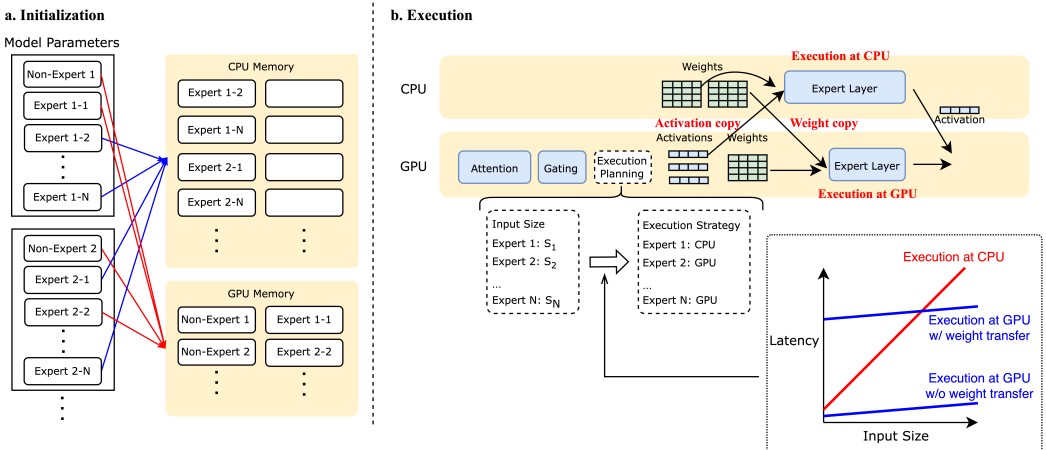

Figure 2: Overview of *Fiddler*. (a) During the initialization phase, the parameters of non-expert layers and a selected subset of expert layers are allocated to the GPU memory as availability permits; the remaining parameters are allocated to the CPU memory. (b) At runtime, *Fiddler* dynamically determines the optimal execution strategy by considering the volume of inputs that activate each expert layer along with the different expected latencies of the CPU and the GPU processing.

reduce the overhead of transferring model parameters, such approaches show suboptimal execution speed for important use cases, such as long prefill or beam search, that are essential for enhanced generation quality (Dong et al., 2022; von Platen, 2023). This is because they do not consider the different batching effects of GPUs and CPUs (Chen, 2023) and the properties of MoE models.

Although model compression techniques like quantization (Frantar & Alistarh, 2023; Zhao et al., 2023) or sparsification (Alizadeh et al., 2023; Tang et al., 2024; Zhu et al., 2025) can reduce the model size and improve inference efficiency, they come with degraded output quality of models, especially when trying to fit large models to a GPU with limited memory capacity (Eliseev & Mazur, 2023). Recently, (Song et al., 2023) proposed to exploit the LLMs sparsity for faster inference with CPU offloading. However, this approach requires the model to use the Rectified Linear Units (ReLU) function for the nonlinear activation. Converting non-ReLU models, common in state-of-the-art LLMs, to ReLU models requires additional training and causes degradation of model quality (Mirzadeh et al., 2023; SparseLLM). For example, Mixtral-8x7B uses the Sigmoid Linear Units (SiLU) function (Elfwing et al., 2018), and only a small portion of values are close to zero. Therefore, it is difficult to exploit the sparsity (a more detailed discussion is given in Appendix B). *Fiddler* can achieve better performance without modifying the model structure or accuracy. We note that *Fiddler* is orthogonal to the compression techniques, and these optimizations could be applied on top of *Fiddler*.

## 3 DESIGN

This section explains the design of *Fiddler*. *Fiddler* is designed for the scenario where GPU memory capacity is insufficient to store all the MoE model parameters. Therefore, the weights of some of the experts are stored in the CPU memory instead of the GPU memory. *Fiddler* finds the optimal execution strategy for such cases, given the expert selection by the input and differing batching behavior of CPUs and GPUs.

### 3.1 OVERVIEW

Figure 2 illustrates the overview of *Fiddler*. In the initialization phase, *Fiddler* allocates the parameters for non-expert layers along with those for a selected subset of expert layers to the GPU memory, as much as the GPU memory capacity permits. *Fiddler* selects those experts to be placed on the GPU memory based on their popularity, which we explain in §3.4. *Fiddler* always allocates the weights of non-expert layers on the GPU memory because they are used for every token, irrespective of expert

choice. The size of non-expert layers is usually not big (*e.g.*, less than 2 billion parameters for the Mixtral-8x7B model), and we assume they fit in the GPU memory in this paper. The parameters of expert layers that do not fit in the GPU memory due to capacity constraints are stored in the CPU memory.

During the execution phase, *Fiddler* carefully assesses and selects the most effective execution strategy. This decision is informed by the number of inputs each expert layer receives and the CPU's and GPU's differing processing latencies. The gating layer of the model determines the number of inputs each expert gets, and the processing latencies can be predicted using the device properties.

*Fiddler* considers the different batching effects of CPUs and GPUs. In processing expert layers, the number of inputs affects the execution latency differently on the CPU and the GPU. Specifically, GPU processing exhibits a relatively stable latency across varying input sizes, which can be attributed to its parallel processing capabilities. These capabilities make the execution latency bounded by the time it takes to load parameters from memory. In contrast, the latency associated with CPU processing tends to scale almost linearly with the input size. This linear increase is due to the CPU's weaker computation capabilities than the GPUs', which makes the latency bounded by the computation part, not the memory movement part. We give a more elaborate analysis in the Appendix A.

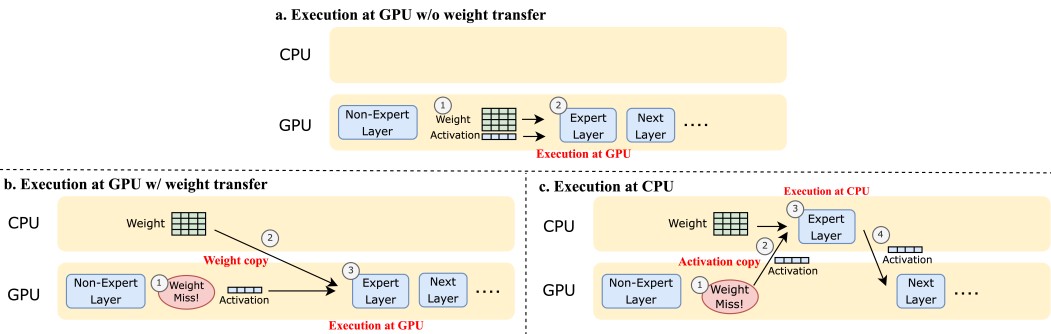

Figure 3: Three different scenarios for the execution of expert layers. When the expert weight is present in GPU memory, the expert layer can be executed at GPU without any data transfer (a). When the expert weight is missing in GPU memory, the expert weight can be copied from CPU memory to GPU memory and executed at GPU (b), or the activation can be copied from GPU memory to CPU memory and executed at CPU (c).

## 3.2 EXECUTION STRATEGIES

After the expert is selected for each input token at the non-expert layer, there are three scenarios for the processing of expert layers, as shown in Figure 3.

If the corresponding weight is present on the GPU memory, the expert layer can be executed on the GPU without any data transfer between the CPU and the GPU (Figure 3 a).

However, as all the model parameters do not fit in the GPU memory, sometimes the expert weights are not present in the GPU memory. In that case, two different strategies exist to execute the expert layer. The first method is to copy the model weight from CPU memory to GPU memory, then execute the expert using the GPU (Figure 3 b). When some expert weights are missing on the GPU memory (①), they are copied from the CPU memory to the GPU memory (②), and then the GPU executes the expert layer (③). Existing offloading systems use this method.

Another approach is to copy the activations from the GPU memory to CPU memory and execute the expert layer on the CPU (Figure 3 c). In this approach, when some expert weights are missing on the GPU memory (①), the activation values are copied from the GPU memory to the CPU memory (②) instead of copying the weights. Then, the computation of the expert layer happens on the CPU (③), and the output activations are copied back to the GPU after the computation finishes (④). A similar method is used by llama.cpp.

The latter two strategies (b. and c. above) have different trade-offs. On the one hand, GPUs have stronger computation ability that is suited for expert processing. Therefore, (b) has an advantage

over (c) regarding computation latency. Moreover, as discussed before, the computation latency of (c) becomes longer as the input size grows due to the different batching effects of the CPUs and the GPUs.

On the other hand, considering the CPU-GPU communication, the method in (b) needs to transfer model weights, while (c) only needs to transfer activation values. As the size of activations (input size $\times$ 4096 for the Mixtral-8x7B) is significantly smaller than the weight size (3 matrices with size $4096 \times 14336$ per expert for the Mixtral-8x7B, consuming more than 300MB with 16-bit precision) for small input sizes, (c) has an advantage in reducing communication overhead.

Overall, (c) is advantageous when the input size to an expert is small, while (b) is better if the input size is above some threshold, even with large communication overhead. When processing long prefills, the input size can reach a thousand. However, in such scenarios, the computation latency of method (c) becomes more prohibitive than the weight transfer latency of (b), making (c) an impractical choice. Consequently, the transfer latency for activation is negligible when (c) is employed. We give a more detailed quantitative analysis in Appendix A.

### 3.3 ALGORITHM

Based on the analysis described above, *Fiddler* serves MoE models in the following way.

**Initialization.** Before starting the inference process, *Fiddler* distributes the model weights between the CPU and GPU memory. First, the weights of non-expert layers are placed on the GPU memory because they are used for every token, irrespective of expert choice. The size of non-expert layers is usually not big (less than 2 billion parameters for the Mixtral-8x7B model), and *Fiddler* assumes they fit in the GPU memory in this paper. Next, *Fiddler* puts a subset of expert layers into the GPU memory. For this, it selects as many experts as the memory capacity permits to maximize the hit rate, *i.e.*, the likelihood an expert's weight is in GPU memory. For the expert selection, we apply an optimization as discussed in §3.4.

We also measure the latency to copy weights and execute experts on either the CPU or the GPU with different input sizes to inform the decision at runtime.

**Execution.** At runtime, *Fiddler* identifies the optimal configurations to execute expert layers across the GPU and CPU. *Fiddler* knows which expert(s) should be used for each token being processed after executing the gating function for each layer. This allows *Fiddler* to learn the input size for each expert. Note that multiple inputs can be processed simultaneously, even for a single request, during the prefill stage or when beam search is utilized.

Based on the input size information, *Fiddler* determines the most efficient execution strategy for distributing workloads across the CPU and GPU. To achieve this, *Fiddler* employs Algorithm 1. The function `is_at_gpu(i, j)` checks whether the weight of expert `j` in the `i`-th layer was placed in the GPU memory at the initialization time. Additionally, `cpu_lat(s)` and `gpu_lat(s)` provide the expected latency for executing an expert on the CPU and GPU, respectively, given an input size of `s`. The function `transfer_lat()` estimates the latency required to transfer an expert's weight from CPU memory to GPU memory.

When executing an expert on a GPU along with weight transfer, the latency is primarily dominated by the time it takes to transfer the expert's weight from the CPU to the GPU memory, which is independent of the batch size. In contrast, executing an expert layer on the CPU demonstrates different behavior: as the number of input tokens increases, the latency also increases. However, the time required to copy activation from the GPU to the CPU is negligible, accounting for less than 1% of the total latency (see Appendix A for more details).

To optimize the processing of the prefill stage, we employ a model where the GPU execution time is considered constant, whereas the CPU execution time is assumed to increase linearly with the number of input tokens. Specifically, for the number of input tokens $s$, `gpu_lat(s)` returns a constant value, while `cpu_lat(s)` returns a value proportional to $s$, multiplied by another constant. These constants are determined in the initialization phase.

---

**Algorithm 1** Expert Execution Strategy

---

```
 1: Inputs:
 2:     n_e: number of experts in one layer
 3:     i: the layer to consider (we consider i-th layer)
 4:     inp_size: array of size of input for each expert
 5: for j = 1 to n_e do
 6:     s ← inp_size[j]
 7:     if s == 0 then
 8:         continue
 9:     end if
10:     if is_at_gpu(i, j) then
11:         // run j-th expert at GPU
12:     else if cpu_lat(s) > gpu_lat(s) + trans_lat() then
13:         // run j-th expert at GPU
14:     else
15:         // run j-th expert at CPU
16:     end if
17: end for
```

---

## 3.4 OPTIMIZATIONS

The best execution performance is achieved when the approach of Figure 3 a is used as frequently as possible, *i.e.*, when the expert weight required by the input is present in GPU memory as frequently as possible. To maximize the likelihood that the required expert is available in GPU memory, we place frequently used experts on the GPU based on offline profiling. For this, we select as many experts as the memory capacity permits in order of popularity to maximize the hit rate, *i.e.*, the likelihood an expert's weight is in GPU memory. We determine the popular experts based on the profile of expert selection using calibration data. We assume this method is enough as the expert selection is known to be based on token characteristics, and the popularity of experts is almost universal across different input domains (Jiang et al., 2024; Xue et al., 2024a). Appendix C discusses expert selection in more detail.

Additionally, we design a specialized computation kernel for expert processing on the CPU using the `AVX512_BF16` instruction set, which is not supported in the native PyTorch implementation (Paszke et al., 2019).

## 4 EVALUATION

We evaluate the performance of *Fiddler* for MoE model inference in resource-constrained settings, where a small number of concurrent queries are given, and the latency is critical for the user experience.

### 4.1 SETUP

**Model and Data.** We use Mixtral-8x7B model (Jiang et al., 2024) with 16-bit precision for the evaluation. This model not only represents the architecture of most of the recent MoE models but is also supported by all the baseline systems, ensuring a fair performance comparison. For the evaluation and calibration data, we use ShareGPT (ShareGPT), a dataset of conversations between humans and chatbots, to model the realistic behavior of expert selection. We pick the subset of conversations randomly. We implement *Fiddler* on top of PyTorch (Paszke et al., 2019). Additionally, we give a sensitivity study on different datasets in §D to show the effectiveness of *Fiddler* in a wider variety of routing behaviors.

**Environments.** We evaluate *Fiddler* on two environments as shown in Table 1. Environment 1 is equipped with weaker CPUs and a GPU than Environment 2 to show the effectiveness of *Fiddler* on a wide range of hardware configurations. None of the environments has enough GPU memory capacity to store all the model parameters. The "Number of Experts on GPU" row shows the maximum

Table 1: Evaluation setups

|  | Environment 1 | Environment 2 |
|---|---|---|
| GPU | Quadro RTX 6000 (NVIDIA, b) | RTX 6000 Ada (NVIDIA, a) |
| GPU Memory | 24576MiB | 49140MiB |
| PCIe | Gen3 x16 (32GB/s) | Gen4 x16 (64GB/s) |
| CPU | Intel(R) Xeon(R) Gold 6126 (48 core) | Intel Xeon Platinum 8480+ (112 core) |
| Number of Experts on GPU | 56/256 | 125/256 |

number of experts that can fit on the GPU memory out of 256 experts (32 layers × 8 experts/layer), giving the memory capacity.

**Baselines.** For baselines, we evaluate DeepSpeed-MII version v0.2.3 (Microsoft), Mixtral-Offloading (Eliseev & Mazur, 2023), and llama.cpp version b2956 (ggml authors, 2023).

For DeepSpeed-MII, we enable ZeRO-Infinity optimization (Rajbhandari et al., 2021) so that it offloads model parameters to the CPU memory and loads them from CPU to GPU dynamically during inference when needed. We enable `pin_memory` in the configuration to use paged-locked CPU memory, which improves performance of read/writes from CPU memory and reduce memory defragmentation.

Mixtral-Offloading originally supports only a quantized version of the Mixtral-8x7B model by default. For a fair comparison, we extend Mixtral-Offloading to support running the original version of the model with 16-bit precision. Mixtral-Offloading provides an `offload_per_layer` parameter to determine how many experts in each expert layer to offload to CPU memory. We set the `offload_per_layer` parameter to 7 for Environment 1 and 5 for Environment 2 as this is the best configuration for the environments we test. For llama.cpp, we set the `ngl` parameters that control the number of layers being executed in the GPU to be 8 for Environment 1 and 16 for Environment 2.

**Metrics.** We evaluate the performance of *Fiddler* against baselines in three different scenarios that serve a single request: ⓐ end-to-end latency [1] with different lengths of input and output tokens, ⓑ prefill processing for the long context input, and ⓒ end-to-end latency of beam search with different widths. These metrics reflect important use cases: long context input is used for in-context learning or retrieval augmented generation (Dong et al., 2022; Gao et al., 2023), and beam search is used for enhanced quality of generated tokens (von Platen, 2023).

We report the inference speed measured by token per second for ⓐ and ⓒ (number of generated tokens divided by the end-to-end latency), and Time To First Token (TTFT) for ⓑ. For the evaluation with $N$ input tokens, we randomly select samples from ShareGPT with $N$ tokens or more of prompt and use the initial $N$ tokens.

For ⓐ, the input length is among [32, 64, 128, 256], and the output length is among [64, 128, 256, 512]. The input length for ⓑ is among [512, 1024, 2048, 4096]. We set the beam search width for ⓒ to be among [4, 8, 12, 16] with an input length of 32 and an output length of 64. For the beam search, we compare *Fiddler* only against llama.cpp as the other baselines do not support beam search inference.

## 4.2 Results

Figure 4 shows the end-to-end performance of four methods in two environments. On average, across all the configurations and environments, *Fiddler* outperforms the best baseline, llama.cpp, *Fiddler* achieves performance that is 1.26 times faster on average across different input/output lengths and environments.

Figure 5 shows the TTFT for the long context prefill. In this case, offloading-based methods (DeepSpeed-MII and (Eliseev & Mazur, 2023)) are better than llama.cpp. Still, *Fiddler* shows better performance than any existing methods, outperforming DeepSpeed-MII by 1.07 times and (Eliseev & Mazur, 2023) by 1.65 times on average across different configurations. Figure 6 shows the end-to-end

---

[1] We define the end-to-end latency to be the time taken from the moment the inference request is received to the generation of the final token, including both the prefill and decode times.

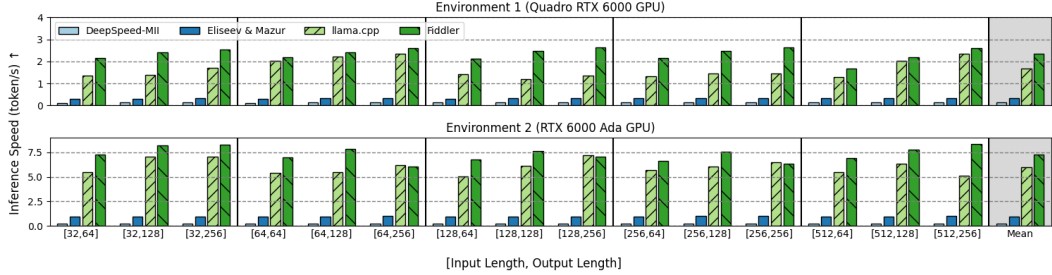

Figure 4: The end-to-end performance comparison by the number of tokens generated per second (scenario ⓐ, higher is better), with 15 different input/output length configurations. The rightmost set of bars shows the average of 15 configurations.

latency of beam search inference with different search widths, compared against llama.cpp. On average, *Fiddler* achieves 11.57 times better performance than llama.cpp.

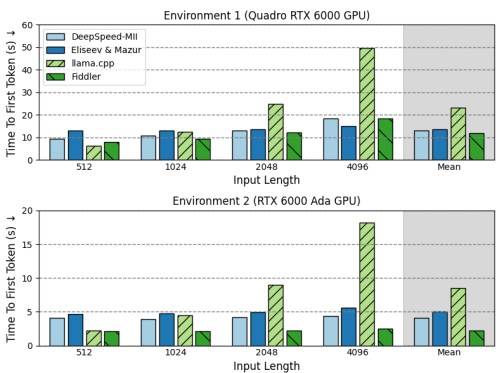

Figure 5: The performance comparison by TTFT (scenario ⓑ, lower is better), with 4 different input length configurations. The rightmost set of bars shows the average of 4 different lengths.

Figure 6: The performance comparison for beam search inference measured by the number of tokens generated per second (scenario ⓒ, higher is better), with input length of 32 and output length of 64. The rightmost set of bars shows the average of 4 beam search widths.

These results show that *Fiddler* performs better in a wide range of applications than existing systems. The benefits primarily come from *Fiddler*'s ability to determine execution strategy dynamically based on batching effects of CPUs and GPUs and place experts based on popularity profile. Notably, while existing systems show different trade-offs (*e.g.*, offloading-based approaches excel in long prefill scenarios and methods like llama.cpp perform well with single batch latency), our system integrates the advantages of both, achieving balanced and efficient results in diverse conditions.

## 5 CONCLUSION

This paper proposes *Fiddler*, a resource-efficient inference system for MoE models with limited GPU resources. *Fiddler* strategically utilizes the heterogeneous computing architecture of CPU and GPU resources by determining the optimal execution strategy. *Fiddler* achieves better performance in all common scenarios for local inference while state-of-the art systems are only optimized for part of them. Our evaluation shows that compared to state-of-the-art systems, *Fiddler* archives 1.26 times speed up in single batch inference, 1.30 times in long prefill processing, and 11.57 times in beam search inference.

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

## A MICROBENCHMARKS

In this section, we show the results of the microbenchmarks. Figure 7 shows the latency of the following workloads:

- `W copy`: Transferring weight of one expert from the CPU memory to the GPU memory
- `A copy`: Transferring one activation from the GPU memory to the CPU memory
- `GPU N`: Executing one expert at GPU with input size $N$ (excluding the time for transferring weight from CPU)
- `CPU N`: Executing one expert at CPU with input size $N$

For each value, we execute the workload 32 times (once for each layer of Mixtral-8x7B) and present the average and standard deviation results.

When tasks are executed on a GPU, the latency for transferring weights from CPU memory to GPU memory is about 2-5 times longer than the actual computation time. The computation latency on the GPU remains largely constant regardless of batch size. An exception occurs in Environment 1 when the batch size is 1, as PyTorch uses different implementations for single-batch and multi-batch scenarios. However, this difference is minor (approximately $10\%$) compared to the overall latency, which includes weight transfer. Therefore, we model GPU latency as a constant in Section 3.3.

On the CPU, execution latency generally increases linearly with the size of the input batch. However, the time needed to transfer activations is negligible (less than $1\%$ of the latency with a single input). Due to this minimal impact, our model in Section 3.3 assumes that CPU latency has a linear relationship with the number of inputs.

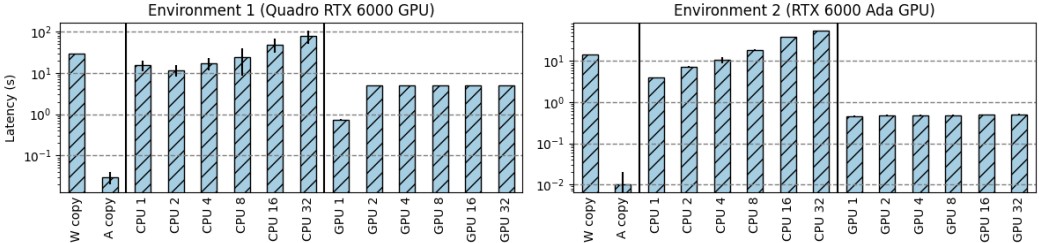

Figure 7: The microbenchmark results measuring the latency of transferring weights or activations between the CPU and GPU, as well as executing an expert layer on either the CPU or GPU with varying input sizes. The y-axis is displayed on a log scale.

## B  SPARSITY ANALYSIS

This section analyzes the sparsity within Mixtral-8x7B models, highlighting the challenges of applying traditional sparsity-based optimization techniques from previous studies (Song et al., 2023; Alizadeh et al., 2023). These methods primarily target LLMs that utilize the ReLU activation function, which nullifies negative inputs and allows for the pruning of channels with consistently zero outputs. This approach leverages the binary nature of ReLU's output—either zero or positive—enabling straightforward identification and elimination of inactive channels, thereby optimizing computational efficiency without sacrificing critical information.

Conversely, state-of-the-art MoE models often use different activation functions, complicating the direct application of these sparsity-exploiting strategies. For instance, Mixtral-8x7B uses SiLU as the activation function. Unlike ReLU, SiLU does not provide a clear threshold of zero for pruning, necessitating a more sophisticated approach to leverage sparsity. Pruning channels that are not sufficiently close to zero could negatively impact the model's output quality.

Table 2 presents an analysis of the absolute values after the SiLU function across the layers of the Mixtral-8x7B model. This analysis is based on data from 100 samples within the ShareGPT dataset (ShareGPT), without differentiating between different experts in identical layers. The data indicates a generally low occurrence of values close to zero. Specifically, for all layers, the proportion of channels with absolute values below 0.001 is less than $2\%$, and for 30 out of the 32 layers, this figure is even below $1\%$. Additionally, in 28 out of 32 layers, fewer than $5\%$ of the values are smaller than 0.01, and in 24 layers, fewer than $30\%$ of the values are under 0.1. Despite variations across layers, these results collectively suggest a significant challenge in harnessing sparsity within this model using approaches from previous works. In contrast, it is reported (Liu et al., 2023) that over $90\%$ of values after the ReLU function are zero for the MLP layers of OPT models (Zhang et al., 2022). Utilizing sparsity within models like Mixtral-8x7B to speed up inference with tolerable quality loss remains an intriguing direction for future research.

Table 2: Distribution of absolute values after SiLU function of Mixtral-8x7B model across all layers. Each cell displays the percentage of values whose absolute value is below a specified threshold.

| Layer | < 0.001 | < 0.01 | < 0.1 | < 1.0 |
|-------|---------|--------|-------|-------|
| 1  | 1.75 | 17.17 | 93.89 | 100.00 |
| 2  | 1.21 | 11.95 | 85.08 | 100.00 |
| 3  | 0.92 | 9.10  | 74.80 | 99.99 |
| 4  | 0.71 | 7.06  | 63.69 | 99.99 |
| 5  | 0.50 | 5.00  | 49.67 | 99.95 |
| 6  | 0.41 | 4.08  | 41.60 | 99.93 |
| 7  | 0.36 | 3.56  | 36.66 | 99.91 |
| 8  | 0.30 | 2.97  | 31.04 | 99.88 |
| 9  | 0.29 | 2.90  | 29.96 | 99.86 |
| 10 | 0.27 | 2.73  | 28.25 | 99.80 |
| 11 | 0.24 | 2.37  | 24.65 | 99.74 |
| 12 | 0.24 | 2.43  | 25.15 | 99.69 |
| 13 | 0.24 | 2.36  | 24.55 | 99.65 |
| 14 | 0.22 | 2.22  | 23.05 | 99.53 |
| 15 | 0.20 | 2.02  | 21.03 | 99.32 |
| 16 | 0.18 | 1.78  | 18.61 | 99.14 |
| 17 | 0.15 | 1.53  | 16.14 | 98.91 |
| 18 | 0.15 | 1.50  | 15.86 | 98.58 |
| 19 | 0.13 | 1.33  | 14.24 | 98.15 |
| 20 | 0.12 | 1.19  | 12.94 | 97.95 |
| 21 | 0.11 | 1.09  | 12.04 | 97.86 |
| 22 | 0.10 | 0.97  | 11.09 | 97.96 |
| 23 | 0.10 | 1.02  | 11.58 | 97.61 |
| 24 | 0.10 | 1.02  | 11.72 | 97.36 |
| 25 | 0.09 | 0.95  | 11.55 | 97.34 |
| 26 | 0.10 | 0.95  | 11.91 | 97.05 |
| 27 | 0.09 | 0.95  | 12.19 | 96.72 |
| 28 | 0.09 | 0.89  | 12.28 | 96.76 |
| 29 | 0.08 | 0.86  | 13.89 | 95.86 |
| 30 | 0.09 | 1.03  | 15.16 | 94.02 |
| 31 | 0.12 | 1.37  | 16.65 | 92.12 |
| 32 | 0.36 | 2.73  | 20.27 | 89.64 |

## C    EXPERT POPULARITY

Figure 8 displays a heat map illustrating the popularity of expert selection within the Mixtral-8x7B model. Similar to the analysis in Appendix B, this profile is generated by running inferences on random samples from the ShareGPT dataset and counting the number of tokens routed to each expert. The color intensity of each cell represents the frequency of expert selection, equivalent to the number of tokens that activated the expert. The value of the most popular expert is normalized to 1, with the popularity of other experts expressed as a ratio relative to this value.

Among the 256 experts, the average value is 0.71, with a standard deviation of 0.08, a 25th percentile of 0.67, and a 75th percentile of 0.76. Although the minimum value is 0.22, only 15 experts have values below 0.6, and 27 experts exceed 0.8, indicating a relatively balanced distribution.

In Environment 1, selecting the 56 most popular experts out of 256 yields a maximum expected hit rate (the likelihood that an expert's weight is available in the GPU memory) of 25.2%, compared to a minimum of 18.7%. Random selection results in an average hit rate of $56/256 = 21.9\%$. In Environment 2, with GPU memory capacity for 125 experts, the expected hit rates for the best, worst, and random selections are 53.0%, 44.6%, and 48.8%, respectively. Therefore, we can conclude that placing popular experts on the GPU could improve the hit rate by approximately 3 to 5 percentage points compared to random placement.

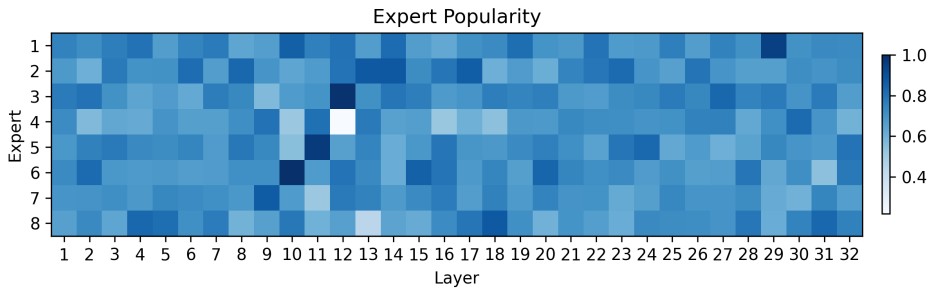

Figure 8: A heat map visualizing expert selection frequency in the Mixtral-8x7B model, using color intensity to represent the frequency, with the most popular expert normalized to 1.

## D  SENSITIVITY STUDY ON DATASET

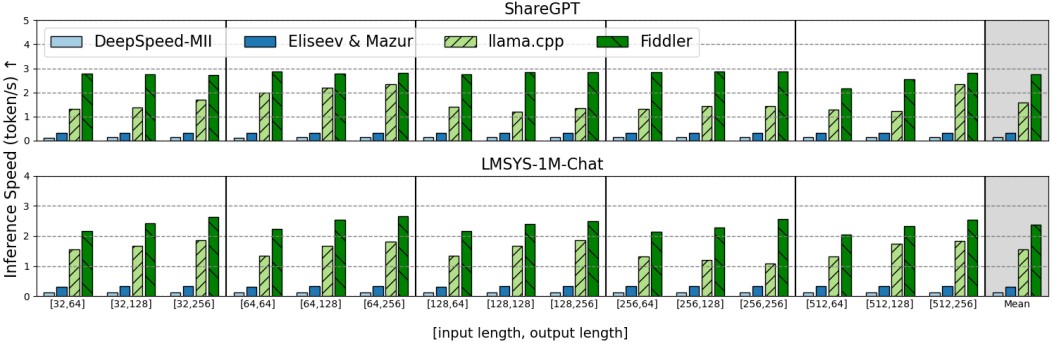

Figure 9: The end-to-end performance comparison by the number of tokens generated per second (same as scenario ⓐ, higher is better), with two different datasets The rightmost set of bars shows the average of 15 configurations.

In this section, we analyze the sensitivity of *Fiddler*'s performance on input datasets since MoE models' routing behavior can be affected by the characteristics of input data distribution. Figure 9 compares the performance of *Fiddler* with ShareGPT (ShareGPT) and LMSYS-Chat-1M datasets (Zheng et al., 2024), both of which are datasets of conversation between humans and chatbots. Aside from the dataset, experimental setups are the same as scenario ⓐ in §4, and we use Environment 1.

On average, *Fiddler* outperforms the state-of-the-art system (llama.cpp) by 1.81 times for the ShareGPT dataset and 1.56 times for the LMSYS dataset. These results show *Fiddler*'s robustness to different distributions of inputs.

## E  APPLICABILITY OF *Fiddler* FOR DIFFERENT MODELS

In the §4, we evaluated the Mixtral-8x7B model because it is the only MoE model that is supported by all of the baselines. However, our system is designed to be model-agnostic within the family of

MoE models. To demonstrate this, Figure 10 presents *Fiddler*'s performance for the Phi-3.5-MoE model (Abdin et al., 2024). We show the comparison against DeepSpeed-MII, since it is the only baseline system that supports this model.

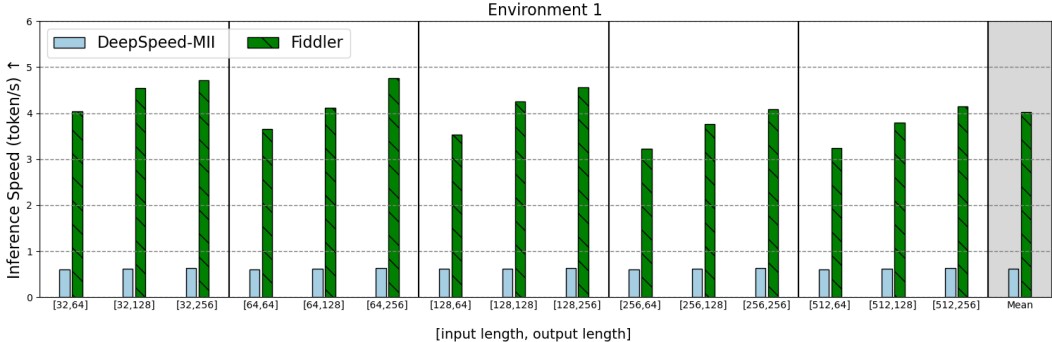

Figure 10: The end-to-end performance comparison of Phi-3.5-MoE model by the number of tokens generated per second (same as scenario ⓐ, higher is better.)

The results are consistent with the Mixtral-8x7B model, and *Fiddler* outperforms DeepSpeed-MII with 6.5 times on average. It shows the applicability of *Fiddler* beyond Mixtral-8x7B model.

## F   BREAKDOWN OF THE LATENCY

Figure 4 shows the performance as measured by the number of generated tokens divided by end-to-end latency. To complement the data, Figure 11 and Figure 12 show the Time To First Token (TTFT) and Inter-Token Latency (ITL) separately. In terms of TTFT, *Fiddler* shows an average of 1.13 times speedup across different input lengths among all baselines in 2 environments. In terms of ITL, *Fiddler* shows an average of 1.43 times speedup across different input and output lengths among all baselines in 2 environments.

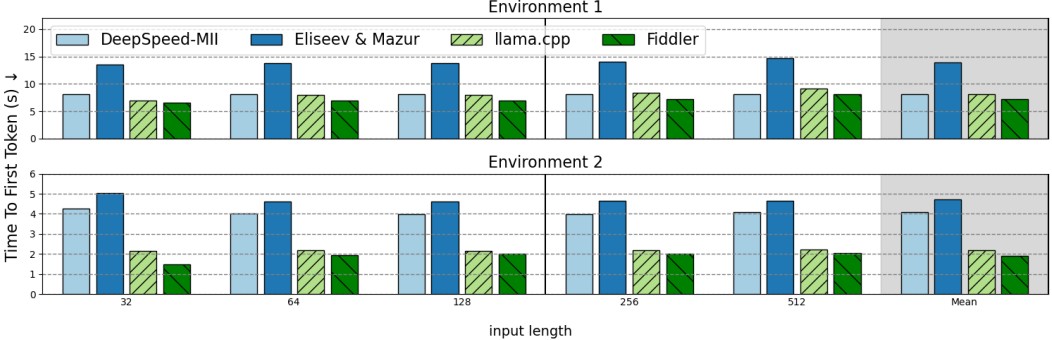

Figure 11: The Time-To-First-Token (TTFT) comparison (lower is better).

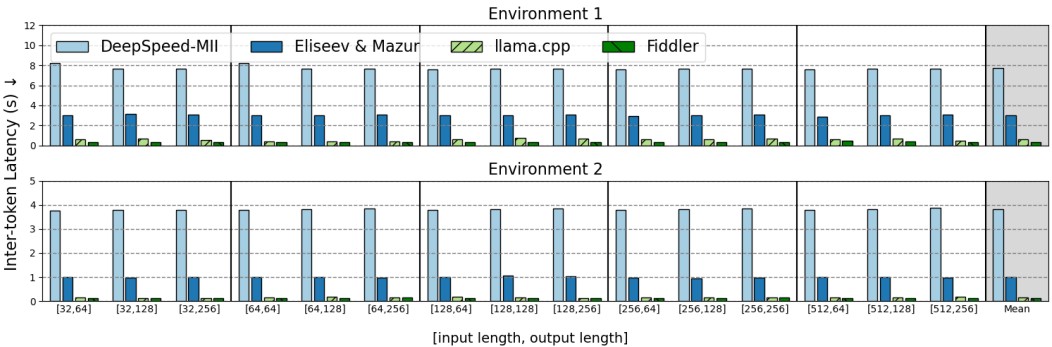

Figure 12: The Inter-Token Latency (ITL) comparison (lower is better).

