# OpenReview forum: "Fiddler: CPU-GPU Orchestration for Fast Inference of Mixture-of-Experts Models"
_ICLR.cc/2025/Conference — ICLR 2025 Poster_

### Official Review · Reviewer_eBnt · 2024-10-28

**Soundness:** 3
**Presentation:** 4
**Contribution:** 3
**Rating:** 8
**Confidence:** 4

**Summary:**

This paper discusses how to optimize the inference performance of MoE in a resource-constrained environment but with heterogeneous hardware. Because the model size is too large to fit into the GPU memory, existing systems offload weights or computations to CPU resources. However, they didn’t consider the MoE model properties or different device characteristics of CPUs and GPUs, leading to suboptimal performance. To optimize the performance, this paper proposes Twiddler that finds the optimal execution strategy using both GPU and CPU resources. Its evaluation demonstrates that Twiddler outperforms existing systems in different applications and hardware settings.

**Strengths:**

+ This paper is well-written and clearly defines the scope.
+ This paper analyzes the fundamental limitations of existing systems for MoE inference in a resource-constrained environment
+ This paper discusses the design space of MoE inference in a resource-constrained environment: 1/ expert weight is on GPU memory; 2/ expert weight is not on GPU and it is copied copied from CPU to GPU; and 3/ expert weight is not on GPU but the activation is copied from GPU to CPU. It then devises a cost model to dynamically determine whether to execute computation on GPU or CPU.
+ This paper evaluates different input and output length on two hardware settings to demonstrate the effectiveness of Twiddler.

**Weaknesses:**

- The used metrics can be further improved: 1/ the definition of end-to-end latency is unclear; 2/ Typically TTFT and inter-token latency (ITL) will be both used as metrics for inference performance evaluation, but only TTFT is used in this paper. Although token per second is used, its definition still mingles with TTFT.
- Only single batch size is used for evaluation. Curious if Twiddler still works for scenarios with batch size over 1, such as 2, 4, and 8
- The performance improvement of Twiddler over the best baselines is marginal. In terms of TTFT, it only outperformed DeepSpeed-MII by 1.07x.

**Questions:**

N/A

---

> ### Author Response · Authors · 2024-11-18
>
> Thank you for your comments.
>
> > The used metrics can be further improved: 1/ the definition of end-to-end latency is unclear; 2/ Typically TTFT and inter-token latency (ITL) will be both used as metrics for inference performance evaluation, but only TTFT is used in this paper. Although token per second is used, its definition still mingles with TTFT.
>
> 1/ We define the end-to-end latency to be the time taken from the moment the inference request is received to the generation of the final token, including both the prefill and decode times. We clarified this point in the footnote of Section 4.1 of the revised PDF.
>
> 2/ We measure end-to-end latency, which includes both the TTFT and ITL. The tables below show the breakdown of TTFT and ITL for Environment 1 of Figure 4.
>
> TTFT (s):
>
> | Input length | Twiddler (ours) | llama.cpp | DeepSpeed-MII | Mixtral-offloading |
> |--------------|----------|-----------|---------------|--------------------|
> | 32           | 6.59     | 7.00      | 8.10          | 13.58              |
> | 64           | 6.90     | 8.03      | 8.10          | 13.72              |
> | 128          | 6.97     | 8.02      | 8.10          | 13.72              |
> | 256          | 7.24     | 8.41      | 8.11          | 14.01              |
> | 512          | 8.10     | 9.10      | 8.14          | 14.71              |
>
> ITL (s):
>
> | Input length, Output length | Twiddler (ours) | llama.cpp | DeepSpeed-MII | Mixtral-offloading |
> |-----------------------------|----------|-----------|---------------|--------------------|
> | 32, 64                      | 0.36     | 0.64      | 8.21          | 3.01               |
> | 32, 128                     | 0.36     | 0.66      | 7.63          | 3.12               |
> | 32, 256                     | 0.37     | 0.56      | 7.66          | 3.07               |
> | 64, 64                      | 0.35     | 0.40      | 8.21          | 3.01               |
> | 64, 128                     | 0.36     | 0.40      | 7.63          | 3.02               |
> | 64, 256                     | 0.36     | 0.40      | 7.66          | 3.07               |
> | 128, 64                     | 0.36     | 0.60      | 7.57          | 3.01               |
> | 128, 128                    | 0.35     | 0.77      | 7.63          | 3.02               |
> | 128, 256                    | 0.35     | 0.70      | 7.66          | 3.07               |
> | 256, 64                     | 0.35     | 0.63      | 7.57          | 2.91               |
> | 256, 128                    | 0.35     | 0.63      | 7.63          | 3.02               |
> | 256, 256                    | 0.35     | 0.66      | 7.66          | 3.07               |
> | 512, 64                     | 0.46     | 0.64      | 7.57          | 2.90               |
> | 512, 128                    | 0.39     | 0.66      | 7.63          | 3.01               |
> | 512, 256                    | 0.37     | 0.50      | 7.66          | 3.07               |
>
> We added these data along with the TTFT and ITL breakdown for Environment 2 to Appendix F of the revised PDF.
>
> > Only single batch size is used for evaluation. Curious if Twiddler still works for scenarios with batch size over 1, such as 2, 4, and 8
>
> Twiddler supports batch inference, but we did not evaluate it because only the single batch inference is typically used in the local setting that we consider in the paper. For multiple batches, the performance is similar to that of beam search inference, because beam search is effectively multi-batch inference at the token level.
>
> > The performance improvement of Twiddler over the best baselines is marginal. In terms of TTFT, it only outperformed DeepSpeed-MII by 1.07x.
>
> Twiddler aims to create a system that performs consistently well across diverse scenarios, unlike existing solutions which often excel in one area while compromising in others. For example, offloading-based systems like DeepSpeed-MII have an advantage in long context inputs (Figure 5), while CPU-based systems like llama.cpp are better in other cases (Figure 4). Our system outperforms all those different state-of-the-art systems.

---

> > ### Comment · Reviewer_eBnt · 2024-11-26
> >
> > I acknowledge the authors' rebuttal and will keep my score

---

### Official Review · Reviewer_GVs3 · 2024-10-30

**Soundness:** 3
**Presentation:** 3
**Contribution:** 3
**Rating:** 6
**Confidence:** 4

**Summary:**

The authors propose a resourceefficient inference system for MoE models with limited GPU resources,which can strategically utilizes CPU and GPU resources by determining the optimal execution strategy. It performs better in all scenarios.Additionally, the  authors design a specialized computation kernel for expert processing on the CPU using the AVX512_BF16 instruction set,

**Strengths:**

1. The algorithm in the experiment achieved good results.Twiddler achieves 1.26 times speed up in single batch inference, 1.30 times in long prefill processing, and 11.57 times in beam search inference.
2. it is good to  analyze the sensitivity of Twiddler’s performance on input datasets to discover whether  MoE models’ routing behavior may be affected by the characteristics of input data distribution.
3. it seems good to  design a specialized computation kernel for expert processing on the CPU using the AVX512_BF16 instruction set

**Weaknesses:**

1. The experiments are a bit sparse and insufficient. There should be experiments where experts have better data from CPU to GPU, which can better explain the reasons for the performance improvement.
2. A dedicated computing core is designed for expert processing on the CPU. However, its impact was not considered during the experiment.
3. More details about the way to find the suitable experts in the GPU rather than CPU are needed, and how the inference can comparable with SOTA inference framework like TensorRT etc.

**Questions:**

1. In section 2.2, when it states, “such approaches show suboptimal performance for important use cases,” what specific metrics are being referred to as suboptimal? Is it in terms of execution time, accuracy, resource utilization, or something else?
2. At runtime, Twiddler identifies the optimal configurations for executing expert layers across the GPU and CPU. Could you provide more details on how Twiddler determines which expert(s) to use for each token after executing the gating function for each layer? What methods are used to obtain the optimal configuration? Are there any experimental studies to support this?
3. Could you explain the concepts of beam search and speculative decoding? How does speculative decoding function within this system? More experimental results would be helpful for clarification.
4. What is the memory usage for this work? How does it compare in terms of memory savings with approaches that do not implement switching?
5. How does this work compare with state-of-the-art LLM inference methods such as vLLM, TensorRT, or other LLM serving frameworks? While I understand some of them may not support the Mixture of Experts (MoE) framework, how does the baseline performance compare?
6. If one expert is too slow, what is the overhead associated with loading this expert into GPU memory? How are the costs related to switching experts between GPU and CPU managed? Is there a specific switching policy in place?
7. The author mentions that a router is used to decide when to use the CPU and GPU for the next token. What is the overhead cost of the router? Additionally, if the tokens are broadcast to each expert, can the proposed approach still function effectively

---

> ### Author Response · Authors · 2024-11-18
>
> Thank you for your comments.
>
> > In section 2.2, when it states, “such approaches show suboptimal performance for important use cases,” what specific metrics are being referred to as suboptimal? Is it in terms of execution time, accuracy, resource utilization, or something else?
>
> We are sorry for the confusion. We meant execution time. Since those methods incur suboptimal scheduling of data movement or computation, their execution speed is worse than our system’s. We modified Section 2.2 of the PDF to clarify this point.
>
> > At runtime, Twiddler identifies the optimal configurations for executing expert layers across the GPU and CPU. Could you provide more details on how Twiddler determines which expert(s) to use for each token after executing the gating function for each layer? What methods are used to obtain the optimal configuration? Are there any experimental studies to support this?
>
> The gating function of the MoE model is responsible for choosing the expert for each token, which is a common design in MoE models. To obtain the optimal configuration, we use Algorithm 1 described in the paper, which achieves better performance than other systems as per our evaluation. In a nutshell, the algorithm estimates the costs (execution time) of different execution strategies and chooses the optimal one.
>
> > Could you explain the concepts of beam search and speculative decoding? How does speculative decoding function within this system? More experimental results would be helpful for clarification.
>
> Beam search is a sampling algorithm that explores the most promising possibilities in a search tree to find the best solution, rather than exhaustively searching every path that token generation can take. Beam search is widely used for enhancing LLM generation quality (Dong et al.,2022; von Platen, 2023).
>
> As for speculative decoding, we did not mention it in this paper. Since most of the speculative decoding methods require training the draft model or finetuning and do not support the Mixtral model specifically, it is difficult to provide experimental results for it. However, the performance would be similar to beam search, because both are processing multiple tokens at a time from the system perspective.
>
> > What is the memory usage for this work? How does it compare in terms of memory savings with approaches that do not implement switching?
>
> Compared to baseline systems, there is no change in total memory usage since our method does not involve compressing the model. Note that existing compression methods (quantization, sparsification, etc.) are complementary and compatible with our system.
>
> > How does this work compare with state-of-the-art LLM inference methods such as vLLM, TensorRT, or other LLM serving frameworks? While I understand some of them may not support the Mixture of Experts (MoE) framework, how does the baseline performance compare?
>
> Those frameworks are GPU-only frameworks that assume access to enough GPU compute and memory, and therefore, they are not directly comparable. The most popular framework for CPU-GPU inference would be llama.cpp and we provide the performance comparison in the paper.
>
> > If one expert is too slow, what is the overhead associated with loading this expert into GPU memory? How are the costs related to switching experts between GPU and CPU managed? Is there a specific switching policy in place?
>
> We give microbenchmark results regarding these overheads in Appendix A. If one expert becomes too slow by processing many tokens as input, the weight transfer cost is smaller than the execution cost on the CPU, so the execution would take place on the GPU. We explain the switching policy in Algorithm 1, which estimates the costs (execution time) of different execution strategies and chooses the optimal one.
>
> > The author mentions that a router is used to decide when to use the CPU and GPU for the next token. What is the overhead cost of the router? Additionally, if the tokens are broadcast to each expert, can the proposed approach still function effectively
>
> The overhead of expert scheduling is minimal, consuming less than 3% of the total execution time. MoE models do not broadcast the tokens to all the experts, but only to select a subset of experts (2 out of 8 for the Mixtral model).

---

> > ### Comment · Reviewer_GVs3 · 2024-11-26
> > **Keep my score after going through the comments**
> >
> > As the title, I would keep my score after going all these comments from different reviewers.

---

### Official Review · Reviewer_jZ6T · 2024-11-04

**Soundness:** 1
**Presentation:** 2
**Contribution:** 1
**Rating:** 5
**Confidence:** 4

**Summary:**

This paper proposes Twiddler, an efficient inference system for MoE models with access to a constrained number of GPUs. Twiddler applied heterogeneous computing mechanisms to leverage the capabilities of both CPUs and GPUs to accelerate inference, given optimal execution strategies. Twiddler claims salient performance improvements in three scenarios, single batch inference, long prefill processing, and beam search inference.

**Strengths:**

The paper systematically approached scenarios where mixed use of CPUs and GPUs can outperform performance of purely relying on GPUs for MoE models.

**Weaknesses:**

Most industry-level inference systems rely predominantly on GPUs for training and inference. At least for current system demands, CPUs are less often adopted in practical heterogeneous clusters, considering the tremendous flops to be performed. Although there may be use cases where communicating weights of small models across CPUs and GPUs may have a slight improvement, this formulation has little practical implications for models of medium and large sizes due to the tremendous computational latencies. Secondly, input size is only but one aspect that contributes to the single batch latency in inference as model capacity also contributes significantly to the total flops. Therefore, this assumption on input size requires refinement. Thirdly, the sequence lengths/widths used in the experiments are not representative enough to demonstrate the generalizability of your formulation. Hence, the paper is recommended for rework.

**Questions:**

Both the formulation and experiments are recommended for rework.

---

> ### Author Response · Authors · 2024-11-18
>
> Thank you for your comments.
>
> > Most industry-level inference systems rely predominantly on GPUs for training and inference. At least for current system demands, CPUs are less often adopted in practical heterogeneous clusters, considering the tremendous flops to be performed. Although there may be use cases where communicating weights of small models across CPUs and GPUs may have a slight improvement, this formulation has little practical implications for models of medium and large sizes due to the tremendous computational latencies.
>
> It is true that GPUs are predominant in massive-scale inference systems in the industry, but there is also fast-growing interest in local settings with less abundant resources. The popularity of projects like llama.cpp (with over 67k stars on GitHub) demonstrates a strong interest in running LLMs on resource-constrained environments. Furthermore, in addition to the studies cited in Section 1, our work aligns with a significant body of recent research focused on efficient LLM inference in such settings [1–3]. Therefore, we believe that the scenario explored in our paper holds substantial practical relevance and impact.
>
> > Secondly, input size is only but one aspect that contributes to the single batch latency in inference as model capacity also contributes significantly to the total flops. Therefore, this assumption on input size requires refinement.
>
> Thanks for the suggestion to explore different factors that might impact total FLOPs. We evaluated the Mixtral model in the paper because it is the only MoE model that is supported by all of the baselines. However, our system is designed to be model-agnostic within the family of MoE models. For example, the following table presents results with the [Phi-3.5-MoE](https://huggingface.co/microsoft/Phi-3.5-MoE-instruct) model, compared against DeepSpeed-MII (the only baseline system that supports this model).
>
> End-to-end latency (token / s)
> | Input length, Output length | Twiddler (ours) | DeepSpeed-MII |
> |-----------------------------|-----------------|---------------|
> | 32,64                       | 4.04            | 0.60          |
> | 32,128                      | 4.55            | 0.62          |
> | 32,256                      | 4.72            | 0.63          |
> | 64,64                       | 3.65            | 0.60          |
> | 64,128                      | 4.12            | 0.62          |
> | 64,256                      | 4.76            | 0.63          |
> | 128,64                      | 3.53            | 0.61          |
> | 128,128                     | 4.25            | 0.62          |
> | 128,256                     | 4.56            | 0.63          |
> | 256,64                      | 3.22            | 0.60          |
> | 256,128                     | 3.76            | 0.62          |
> | 256,256                     | 4.08            | 0.63          |
> | 512,64                      | 3.24            | 0.60          |
> | 512,128                     | 3.80            | 0.62          |
> | 512,256                     | 4.15            | 0.63          |
>
> The results are consistent with the Mixtral model, and Twiddler outperforms DeepSpeed-MII with 6.5 times on average.
> We added these results to the Appendix E of the revised PDF.
>
> > Thirdly, the sequence lengths/widths used in the experiments are not representative enough to demonstrate the generalizability of your formulation.
>
> The following table shows the average token length of ShareGPT and LMSYS-Chat-1M, two datasets of conversations between humans and Chatbots that we used in the paper, which are widely adopted by the community with more than 10,000 monthly downloads.
>
> | Dataset       | Mean Input Length | Mean Output Length |
> |---------------|-------------------|--------------------|
> | ShareGPT      | 246               | 322                |
> | LMSYS-Chat-1M | 102               | 222                |
>
> Therefore, we believe the input/output lengths we used in the paper (between 32 and 512; Figure 4) are representative of real-world use cases.
>
> As for beam width, closely related work uses widths similar to ours. For instance, some studies used a maximum of 16 [4], others tested 2, 4, and 6 [5], some used 4 or 8 [6], and others went up to 20 [7]. Our setup, which uses a beam width between 4 and 16, aligns with these examples. Therefore, we believe our choice about beam width is representative.

---

> ### Author Response · Authors · 2024-11-18
>
> ## References
> [1] Zhenliang Xue, Yixin Song, Zeyu Mi, Le Chen, Yubin Xia, and Haibo Chen. "PowerInfer-2: Fast Large Language Model Inference on a Smartphone." arXiv preprint arXiv:2406.06282 (2024).
>
> [2] Wangsong Yin, Mengwei Xu, Yuanchun Li, and Xuanzhe Liu. "Llm as a system service on mobile devices." arXiv preprint arXiv:2403.11805 (2024).
>
> [3] Haihao Shen, Hanwen Chang, Bo Dong, Yu Luo, and Hengyu Meng. "Efficient llm inference on cpus." arXiv preprint arXiv:2311.00502 (2023).
>
> [4] Charlie Snell, Jaehoon Lee, Kelvin Xu, and Aviral Kumar. "Scaling llm test-time compute optimally can be more effective than scaling model parameters." arXiv preprint arXiv:2408.03314 (2024).
>
> [5] Woosuk Kwon,  Zhuohan Li, Siyuan Zhuang, Ying Sheng, Lianmin Zheng, Cody Hao Yu, Joseph Gonzalez, Hao Zhang, and Ion Stoica. "Efficient memory management for large language model serving with pagedattention." In Proceedings of the 29th Symposium on Operating Systems Principles, pp. 611-626. 2023.
>
> [6] Chufan Shi, Haoran Yang, Deng Cai, Zhisong Zhang, Yifan Wang, Yujiu Yang, and Wai Lam. 2024. A Thorough Examination of Decoding Methods in the Era of LLMs. In Proceedings of the 2024 Conference on Empirical Methods in Natural Language Processing
>
> [7] Yuu Jinnai, Tetsuro Morimura, and Ukyo Honda. "On the depth between beam search and exhaustive search for text generation." arXiv preprint arXiv:2308.13696 (2023).

---

> ### Author Response · Authors · 2024-11-27
>
> Dear Reviewer jZ6T,
>
> Thank you once again for dedicating your time and effort to reviewing our submission. As the end of the discussion period is approaching, we wanted to inquire if there are any remaining questions or feedback from your side. We hope that all the points raised in your feedback have been adequately addressed, but please let us know if anything requires further clarification. We are happy to provide further details and explanations as well as provide new experimental data if time allows.

---

### Author Response · Authors · 2024-11-18

We would like to thank all the reviewers for their time and effort.
Based on the feedback, we modified the following points in the paper. We marked all the changed parts in blue characters.
- Based on the feedback from Reviewer GVs3, we clarified that existing systems show suboptimal performance in terms of execution time in Section 2.2.
- Based on the feedback from Reviewer eBnt, we clarified the definition of end-to-end latency in Section 4.1.
- Based on the feedback from Reviewer jZ6T, we added Appendix E, which shows the performance of Twiddler for another MoE model (Phi-3.5-MoE).
- Based on the feedback from Reviewer eBnt, we added Appendix F, which shows the breakdown of TTFT (time to first token) and ITL (inter-token latency) for the data in Figure 4.

---

### Meta-Review · Area_Chair_Q1TX · 2024-12-11

**Metareview:**

The submission proposes optimization of mixture of experts models where GPU resources are constrained and some processing can be performed on CPUs.  The incorporation of hardware constraints in LLM inference is important, and the reviewers were mostly appreciative of these contributions.  A number of clarifications and additional results were developed during the response period, which can be used to improve the camera ready version.

**Additional Comments On Reviewer Discussion:**

The authors were responsive to reviewer concerns.  One reviewer was not responsive in the discussion period, and did not respond to the authors.  This reviewer was also the only one who indicated the submission was marginally below the threshold.

---

> ### Public Comment · ~Keisuke_Kamahori1 · 2025-02-27
> **Changes to the paper title for the camera ready version**
>
> We have restored the title to its original version, as it appears in our public preprint. This change reverses a temporary modification made during the review process to preserve double-blind anonymity.

---

### Decision · Program_Chairs · 2025-01-22

Accept (Poster)